# Aldehyde Dehydrogenase and Aldo-Keto Reductase Enzymes: Basic Concepts and Emerging Roles in Diabetic Retinopathy

**DOI:** 10.3390/antiox12071466

**Published:** 2023-07-21

**Authors:** Burak Mugdat Karan, Karis Little, Josy Augustine, Alan W. Stitt, Tim M. Curtis

**Affiliations:** Wellcome-Wolfson Institute for Experimental Medicine, Queen’s University Belfast, Belfast BT7 1NN, UK; bkaran01@qub.ac.uk (B.M.K.); k.little@qub.ac.uk (K.L.); j.augustine@qub.ac.uk (J.A.)

**Keywords:** diabetic retinopathy, oxidative stress, lipid peroxidation, aldehyde dehydrogenase, aldo-keto reductase

## Abstract

Diabetic retinopathy (DR) is a complication of diabetes mellitus that can lead to vision loss and blindness. It is driven by various biochemical processes and molecular mechanisms, including lipid peroxidation and disrupted aldehyde metabolism, which contributes to retinal tissue damage and the progression of the disease. The elimination and processing of aldehydes in the retina rely on the crucial role played by aldehyde dehydrogenase (ALDH) and aldo-keto reductase (AKR) enzymes. This review article investigates the impact of oxidative stress, lipid-derived aldehydes, and advanced lipoxidation end products (ALEs) on the advancement of DR. It also provides an overview of the ALDH and AKR enzymes expressed in the retina, emphasizing their growing importance in DR. Understanding the relationship between aldehyde metabolism and DR could guide innovative therapeutic strategies to protect the retina and preserve vision in diabetic patients. This review, therefore, also explores various approaches, such as gene therapy and pharmacological compounds that have the potential to augment the expression and activity of ALDH and AKR enzymes, underscoring their potential as effective treatment options for DR.

## 1. Diabetic Retinopathy (DR)

Diabetic retinopathy (DR) is a progressive disease that poses a serious threat to vision [1,2,3]. Almost all patients affected by type 1 diabetes (T1D) and over 60% of those with type 2 diabetes (T2D) develop DR within 20 years of diabetes diagnosis [4]. The global prevalence of DR among diabetic patients is approximately 28%, and this rate increases to 47% after the age of 60, with no variation between genders in its incidence [5]. As the number of people with diabetes continues to rise, the prevalence of DR is also increasing, placing an ever-growing burden on healthcare systems worldwide [6].

The etiology of DR is strongly associated with hyperglycemia and other factors commonly found in individuals with diabetes, such as hypertension and dyslipidemia [7,8,9]. These factors can lead to a range of pathological events, many of which can be observed and graded in the fundus as lesions associated with retinal microvasculature. Consequently, DR has been historically categorized as a microvascular disease and is broadly classified into distinct progressive stages. The early stage, known as non-proliferative diabetic retinopathy, is characterized by specific effects in the retinal vasculature, including increased vascular permeability, capillary occlusion, altered blood flow, the loss of pericytes, and disruption of endothelial cell tight junctions [2]. Even in the absence of visual symptoms, fundus photography can reveal retinal pathologies such as microaneurysms, hemorrhages, and hard exudates at this stage [2]. The more advanced stage of proliferative diabetic retinopathy (PDR) is characterized by the development of new blood vessels. PDR patients may experience significant visual impairment caused by vitreous hemorrhage or tractional retinal detachment [3]. 

Diabetic macular edema (DME) is the most common cause of vision loss in patients with DR [10]. It is characterized by the swelling or thickening of the macula resulting from the accumulation of fluid beneath the retina and within the retinal layers. This fluid buildup occurs due to the breakdown of the inner and outer blood–retinal barriers (BRBs) [11]. Although DME can occur at any stage of DR, it is most prevalent in the later phases, following progressive vascular and neural damage [12]. DME can be categorized as either ‘central’ or ‘decentralized’, depending on whether it affects the fovea, the central region of the retina, which is responsible for visual acuity [12].

Presently, there are no treatments for the early stages of DR, and the recommended approach is to regulate blood glucose levels and lipid profiles while maintaining proper control over blood pressure [13]. The initial damage caused by diabetes to the retina is not addressed by currently available treatments for PDR or DME, such as laser photocoagulation or intravitreal injections of corticosteroids or anti-Vascular Endothelial Growth Factor (VEGF) therapies [14]. Additionally, clinical trials and real-world evidence indicate that anti-VEGF agents for DR are not universally effective, and there are concerns about the long-term effects of these drugs, including the degeneration of retinal and choroidal blood vessels, as well as neural retina degeneration [15].

Although the clinical classification of DR primarily revolves around vascular pathologies, recent research has provided significant evidence indicating the involvement of the retinal neurovascular unit (NVU) in the development of this disease [16]. The NVU represents a complex network of interactions involving vascular, neuronal, glial, and immune cells. It encompasses the inner BRB and is significantly influenced by the effects of diabetes [17] (Figure 1). Within the retina, various types of neurons (retinal ganglion cells [RGCs], amacrine cells, bipolar cells, photoreceptors, and horizontal cells), glial cells (Müller cells, microglia, and astrocytes), professional immune cells (microglia and perivascular macrophages), and vascular cells (endothelial cells and pericytes) collaborate to maintain the integrity of the inner BRB and regulate blood flow in response to metabolic demands [18]. The diabetes-related disruption of the NVU is characterized by reactive gliosis, diminished retinal neuronal function, and neuronal-cell apoptosis, which can occur before overt microangiopathy in experimental models of DR and in the retinas of people with diabetes [3,17,19]. 

Diabetes causes the breakdown of the inner BRB through the secretion of various substances by glial and immune cells within the NVU, including VEGF, pro-inflammatory cytokines (such as interleukin-1 beta [IL-1β], tumor necrosis factor-alpha [TNF-α], interleukin-6 [IL-6], and monocyte chemoattractant protein-1 [MCP-1]), as well as components of the complement system [20]. The dysregulation of the NVU not only leads to heightened vascular leakage but also disrupts blood flow regulation and causes the thickening of vascular basement membranes. The thickening of these membranes interferes with intercellular communication, affecting survival signaling between the vascular component cells and their interaction with other perivascular elements of the retinal NVU (as reviewed by [1,2]).

In-depth histological analyses of post-mortem retinas obtained from diabetic patients and animal models have highlighted the presence of neurodegeneration as a crucial component of DR. As diabetes progresses, the loss of RGCs, amacrine cells, and even photoreceptors has been observed [21,22,23]. Neurodegeneration in DR is believed to occur through a number of mechanisms that are primarily associated with the disruption of Müller glia, which are the major glial cells in the retina. Müller glia play a crucial role in providing neuroprotective factors such as VEGF and brain-derived neurotrophic factor (BDNF) [24,25,26]. However, in the diabetic retina, the secretion of these factors can be altered [27,28,29]. Additionally, Müller glia exhibit disturbances in their ability to regulate glutamate and potassium (K^+^) in the extracellular space, which has been linked to retinal excitotoxicity [30].

Damage to the retinal NVU during diabetes has been strongly linked with the initiation of oxidative stress, leading to the generation of lipid aldehydes and lipoxidation end-products (ALEs) [31]. The retina, characterized by its rich content of polyunsaturated fatty acids and its high uptake of glucose and oxygen, is particularly vulnerable to oxidative damage [32]. Extensive research has demonstrated that diabetes amplifies oxidative stress and promotes the formation of lipid aldehydes within the retinal NVU, highlighting these processes as significant metabolic abnormalities that are implicated in the development of DR. Understanding the roles of oxidative stress and lipid aldehydes in the pathogenesis of DR presents novel therapeutic opportunities for targeting these mechanisms.

This review aims to delve into the roles of oxidative stress, lipid aldehydes, and ALEs in the pathogenesis of DR while also exploring the therapeutic potential of targeting aldehyde detoxification. The core focus of these discussions center around aldehyde detoxifying enzymes, placing particular emphasis on aldehyde dehydrogenase (ALDH) and aldo-keto reductase (AKR) enzymes. These enzymes exhibit a robust expression in the cells of the retinal NVU and present great potential in effectively addressing the detoxification of aldehydes within the context of DR.

## 2. Reactive Oxygen Species (ROS), Lipid Aldehydes and ALEs

The term “oxidative stress” denotes a state of imbalance between the generation of reactive oxygen species (ROS), such as superoxide ions, hydrogen peroxide, and hydroxyl radicals, and the antioxidant defense mechanisms of the body. ROS are byproducts of cellular metabolism, primarily originating from processes such as mitochondrial oxidative phosphorylation and the activity of the nicotinamide adenine dinucleotide phosphate-oxidase (NOX) system (reviewed by [33,34]). Cells employ antioxidant defensive systems based on enzymatic components, such as superoxide dismutase (SOD), catalase (CAT), glutathione peroxidase (GPx), glutathione transferase, ceruloplasmin, and hemoxygenase to protect against ROS-induced cellular damage [35,36]. Alongside these enzymatic antioxidants, there are several non-enzymatic antioxidants that also play a significant role, including glutathione (GSH), vitamin E, and vitamin C [37]. 

In diabetes, the retina experiences a significant increase in the production of ROS through various mechanisms. These mechanisms include the autoxidation of glucose, the heightened activity of NADPH oxidases, and mitochondrial dysfunction [38,39,40,41,42]. Additionally, diabetes disrupts several metabolic pathways in the retina (Figure 2), such as the polyol pathway, hexosamine pathway, the formation of AGEs, and the protein kinase C (PKC) pathway. These pathways contribute to oxidative stress and the accumulation of ROS [43,44]. Furthermore, mitochondrial dysfunction and elevated ROS levels contribute to the initiation of endoplasmic reticulum (ER) stress [45,46], while disrupted autophagy impairs retinal antioxidant mechanisms, leading to a detrimental cycle of escalating ROS production and oxidative stress [47,48]. To exacerbate the complexity of the situation, the compromised functionality of normal antioxidant defense mechanisms in the retina further amplifies the generation of ROS [49]. 

The factors mentioned above ultimately disturb the equilibrium between ROS production and clearance, resulting in elevated levels of oxidative stress in the diabetic retina (Figure 2). This heightened oxidative stress contributes to an increase in lipid peroxidation and the formation of ALEs, which are explored further in the following sections.

### 2.1. Lipid Peroxidation and ALEs

While ROS can directly impact cellular function and result in cell death, it is increasingly recognized that many harmful effects of oxidative stress stem from ROS-induced lipid peroxidation reactions [50,51]. Lipid peroxidation refers to the metabolic process in which ROS oxidatively degrade lipids, generating lipid aldehydes [52]. It is crucial to note that lipid aldehydes remain highly reactive and can exacerbate oxidative stress by reacting with membrane phospholipids, both within and outside cells [53]. Unlike ROS, lipid aldehydes are relatively stable and can propagate oxidative damage by diffusing sites distant from their origin [53,54]. These aldehydes can covalently bind to nucleophilic side chains of cysteine (Cys), histidine (His), and lysine (Lys) residues in proteins, modifying their structure and function. The resulting chemical adducts are known as ALEs [50,51,53,54]. The adduction of proteins by ALEs adversely affects cell function and survival by influencing the activity of targeted proteins or triggering their rapid degradation through the proteasomal pathway [50]. Furthermore, ROS and reactive aldehydes have the potential to disturb gene expression and interfere with normal intercellular communication processes [31,55,56,57].

Lipid aldehydes and ALEs have been strongly implicated in the pathogenesis of DR [31,58,59]. In the streptozotocin (STZ)-induced rat model of type 1 diabetes (T1D), associations have been observed between lipid peroxidation products in the bloodstream and retinal changes [60,61]. Additionally, elevated levels of lipid hydroperoxides (LOOH) have been detected in the retinas of diabetic humans and STZ-induced diabetic rats [61,62]. Several studies have investigated the potential connection between specific lipid peroxidation products, such as acrolein (ACR), 4-hydroxy-2-nonenal (4-HNE), 4-hydroxyhexenal (4-HHE), malondialdehyde (MDA), and glyoxal (GO), and the development of DR. Below we discuss the experimental and clinical evidence supporting the involvement of these specific lipid aldehydes and their ALEs in the initiation and progression of DR [59]. 

### 2.2. α,β-Unsaturated Aldehydes

#### 2.2.1. Acrolein

ACR, as an extremely reactive α,β-unsaturated aldehyde, exerts its detrimental effect by preferentially binding to Cys, Lys, and His residues, leading to dysfunction of numerous proteins [63,64]. In addition to the peroxidation of unsaturated fatty acids, ACR can be endogenously generated through polyamine metabolism and the myeloperoxidase system present in neutrophils [65]. The enzymes spermine oxidase (SMOX) and acetylpolyamine oxidase (APAO) play a key role in ACR synthesis via the polyamine pathway [66,67]. Neutrophils employ the myeloperoxidase system to combat invading pathogens like viruses and bacteria. Through the 2-hydroxypropanal pathway, neutrophil-generated hypochlorous acid (HOCl) reacts with threonine, leading to ACR production. This mechanism is thought to contribute to long-term tissue damage at sites of inflammation [68]. Among the various adducts formed by ACR-conjugated proteins, the Lys adduct exhibits notable stability. An important example is the Nε-(3-formyl-3,4-dehydropiperidino) lysine adduct (FDP-Lys), which differs from most ALEs as it is considered a reactive intermediate that is capable of covalently binding to thiols, including glutathione (GSH), through its electrophilic carbonyl group. The binding of FDP-Lys to GSH results in a reduction in intracellular GSH levels, depleting the body’s capacity to counteract oxidative stress [69]. 

Our laboratory’s early studies uncovered the significance of FDP-Lys accumulation on retinal Müller cell and neuronal proteins in the development of DR [70,71]. Within a few months of STZ-induced type 1 diabetes, increased levels of FDP-Lys were observed in Müller cells, and as diabetes progressed, this adduct accumulated in retinal ganglion cells (RGCs) and neurons of the inner nuclear layer (INL) [71]. Other research groups have also supported our findings by demonstrating elevated FDP-Lys levels in the vitreous and fibrovascular membranes of patients with PDR [72]. To explore the pathophysiological relevance of FDP-Lys accumulation in DR, we conducted in vitro studies using human Müller cell cultures. These investigations revealed that the accumulation of this adduct triggered Müller cell dysfunction akin to that observed in DR, including oxidative stress induction, the upregulation of pro-inflammatory factors, and dysregulation of K^+^ transport mechanisms [71,73]. In more recent endeavors, we have conducted proof-of-concept studies using the ACR sequestering drug, 2-hydrazino-4,6-dimethylpyrimidine (2-HDP), to assess the potential of blocking FDP-Lys accumulation as a therapeutic strategy for the in vivo treatment of DR. These studies have yielded encouraging results, demonstrating the effectiveness of systemic administration of 2-HDP in preventing NVU dysfunction and retinal pathology in diabetic rats [73]. 

The mechanisms underlying the increased production of ACR and the accumulation of FDP-Lys in the diabetic retina are not yet fully understood. In the vitreous of patients with PDR, the concentration of the polyamine spermine, a precursor of ACR in the polyamine catabolic pathway, was found to be 15 times higher than in control subjects [74]. In a mouse model of diabetes, treatment with MDL 72,527, a dual inhibitor of SMOX and APAO [75], significantly reduced the accumulation of retinal FDP-Lys and prevented neurophysiological dysfunction and neurodegeneration [76]. These findings suggested that both lipoxidation and the dysregulation of the polyamine system could contribute to the buildup of FDP-Lys in the diabetic retina. Notably, our group’s research also indicated that diabetes impaired the retina’s ability to detoxify ACR, which is discussed in more detail later in this review. This impairment may also contribute to the increased accumulation of ACR and the formation of FDP-Lys adducts in the diabetic retina. Overall, the available data strongly suggest that the increased production of ACR and the accumulation of FDP-Lys play a crucial role in the pathogenesis of DR. However, further research is needed to better understand the key sources and underlying mechanisms involved in the production and accumulation of these molecules.

#### 2.2.2. 4-HNE and 4-HHE

One noteworthy cytotoxic compound that emerges during the peroxidation of ω-6 polyunsaturated fatty acids, like arachidonic acid or linoleic acid, is 4-HNE [77]. 4-HNE primarily forms adducts with thiol groups of proteins through 1,2- and 1,4-Michael addition reactions. These adducts subsequently undergo cyclization, leading to the formation of cyclic hemiacetals [78]. Due to its high reactivity with proteins and other macromolecules, 4-HNE is recognized as an extremely toxic byproduct of lipid peroxidation [79]. Nonetheless, cellular enzymatic detoxification mechanisms are capable of swiftly metabolizing 4-HNE (see Section 3 below), facilitating its extensive elimination [80]. As a result, only a small fraction of 4-HNE manages to escape cellular detoxification processes and interact with biomolecules [81]. 

The peroxidation of n-3 polyunsaturated fatty acids, such as docosahexaenoic acid and eicosapentaenoic acid, gives rise to 4-HHE. Structurally similar to 4-HNE, this 4-hydroxyl-2-alkenal forms stable ALE adducts with specific residues (His, Cys, or Lys) on proteins [82]. While sharing some properties, it is worth noting significant distinctions between 4-HHE and 4-HNE. In experiments conducted on cortical neurons, for example, it has been demonstrated that these aldehydes target different proteins within the cell [83]. Additionally, they are detoxified by distinct metabolic pathways [83,84]. Despite 4-HHE’s ability to induce cell dysfunction and cell death [85], it is generally considered less toxic than 4-HNE due to its lower lipophilicity and reduced chemical reactivity [86]. 

Clinical research has demonstrated that diabetic patients with retinopathy exhibit elevated levels of 4-HNE in their blood compared to diabetic patients without retinopathy and healthy controls [87,88]. This finding was supported by studies in diabetic rats, which also showed an increase in 4-HNE and 4-HNE-derived ALEs in the retinas, further indicating its potential involvement in the development of DR [62,89]. 4-HNE is known to activate the WNT signaling pathway and stabilize the WNT co-receptor LRP6, which is implicated in DR [90]. Furthermore, 4-HNE has been associated with changes in retinal perfusion during DR. It has been observed that 4-HNE reduces the activity of Ca^2+^-activated K^+^ (BK) channels [91], which play a key role in retinal arteriolar vasodilation [91]. This impairment of the BKCa pathway leads to a decrease in retinal vasodilator responses to β2-adrenoceptor agonists, indicating that 4-HNE directly impacts retinal blood vessels and contributes to the vascular dysfunction observed in the diabetic retina [91]. Exposure to 4-HNE has also been found to induce endoplasmic reticulum stress, mitochondrial dysfunction, and apoptosis in human retinal capillary pericytes and in vitro cultures of Müller glia [92,93]. In summary, these findings suggest that 4-HNE could contribute to the development of DR by activating the WNT signaling pathway, affecting retinal hemodynamics, and disrupting various cellular processes in the retina. 

Compared to 4-HNE, the role of 4-HHE in the development of DR has received relatively limited attention. Our previous findings indicated that there was no substantial evidence to suggest a significant increase in protein modifications by 4-HHE in the retinas of diabetic rats after four months of disease duration [71]. Further research is needed to explore the potential contributions of 4-HHE in the development and progression of DR, considering the limited research conducted on this specific aldehyde.

#### 2.2.3. Dialdehydes

##### MDA

MDA is produced when polyunsaturated fatty acids are attacked by free radicals, specifically targeting the carbon–carbon double bonds [94]. This attack causes the release of water and the formation of unsaturated lipid radicals. As additional oxygen is captured, lipid hydroperoxides and peroxyl radicals are formed. The presence of a cis–double bond adjacent to the peroxyl group allows for the peroxyl radical to undergo cyclization. During this process, intermediate free radicals are generated, which can further undergo cyclization to form bicyclic endoperoxides. These endoperoxides share structural similarities with prostaglandins and can be cleaved to produce MDA [95,96].

MDA has garnered significant research interest in the field of DR due to its widespread use as a reliable marker for lipid peroxidation [97]. Studies have consistently shown elevated MDA levels in the blood of patients with DR [98,99]. To evaluate the balance between lipid peroxidation stress and cellular antioxidant capacity, researchers have normalized serum MDA levels to superoxide dismutase (SOD). Notably, DR patients exhibited significantly lower SOD/MDA ratios compared to diabetic patients without DR [100]. Some evidence has suggested that individuals with diabetes may have higher MDA levels, specifically in their eyes. For instance, an analysis using high-performance liquid chromatography (HPLC) of sub-retinal fluid from individuals with retinal detachment revealed elevated MDA concentrations in those with diabetes [101]. Additionally, MDA concentrations were found to be increased in the lenses of diabetic patients undergoing cataract surgery [102]. Nonetheless, further investigations are needed to determine the potential role of MDA in the development of DR and its usefulness as a biomarker for detecting the onset of this disorder.

##### Glyoxal

GO, the simplest dialdehyde, is generated through the peroxidation of lipids that are derived from arachidonic acid (AA) and linoleic acid (LA). However, GO exhibits inherent instability as an aldehyde and readily undergoes hydration and polymerization when dissolved in a solution [103]. Consequently, GO acts as a natural fixative, leading to the alkylation of biomolecules and the formation of crosslinks. When GO reacts with protein residues, modifications occur in both Lys and arginine (Arg), giving rise to the formation of N ϵ-(carboxymethyl) lysine (CML) and N ω-carboxymethyl arginine (CMA), respectively [104].

The adducts resulting from the interaction of GO are commonly known as, as they can be formed through glycation or lipoxidation reactions [105,106,107]. Comprehensive investigations have been carried out to measure N ϵ-(carboxymethyl) lysine (CML) levels in the retinas of post-mortem human donor eyes and the experimental rodent models of diabetes [58,108,109,110]. These studies have consistently shown a significant increase in the accumulation of CML-modified proteins in the vascular, glial, and neural tissues of the diabetic retina, leading to detrimental effects on their normal retinal function and cell survival [108,111]. Clinical studies have further indicated that CML levels in memory T cells [112] and skin biopsies [113] could potentially serve as predictive markers for the progression of DR in the future.

## 3. Aldehyde Detoxification in the Retina in Health and Diabetes 

Cells possess protective mechanisms for the rapid metabolism and detoxification of lipid aldehydes, which were formed during lipid peroxidation. These defense mechanisms encompass several processes, such as the oxidation of lipid aldehydes to acids, their reduction to alcohol, and conjugation with vital cellular nucleophiles like ascorbic acid, GSH, and carnosine [114]. Two primary enzymatic pathways play a crucial role in the in vivo detoxification of lipid aldehydes: the aldehyde dehydrogenase (ALDH) and aldo-keto reductase (AKR) superfamilies. The ALDH superfamily encompasses NAD(P+)-dependent enzymes that irreversibly convert aldehydes into carboxylic acids [115]. Conversely, AKRs form a diverse group of enzymes that exhibit catalytic activity when modifying various substrates. These enzymes operate as NADPH-dependent oxidoreductases, facilitating a reduction in distinct aldehydes and ketones into primary and secondary alcohols, respectively [116]. 

In previous work [117], we conducted mRNA expression screening of ALDH and AKR enzymes in rat retina. Initially, we analyzed retinas from control Sprague Dawley rats, focusing on genes known to participate in lipid aldehyde detoxification. The results revealed the strong mRNA expression of several ALDH enzymes, including ALDH1a1, ALDH2, ALDH3a1, ALDH3a2, ALDH9a1, and ALDH18a1. Conversely, a limited subset of AKR enzymes showed an expression, specifically, of AKR1b1, AKR1c19, and AKR7a2. Additionally, we explored the impact of diabetes on the expression of ALDH and AKR enzymes in the rat retina. Our objective was to determine whether diabetes-related impairment of these enzymes contributes to the accumulation of lipid aldehydes and ALEs in the diabetic retina. In the following section, we provide an overview of the substrate specificity localization patterns, and functions of these enzymes in the retina based on the available information. We also highlight the influence of diabetes on their expression and their potential contribution to the development of DR. A summary of the key characteristics of these enzymes is presented in Table 1.

### 3.1. ALDH1A1

ALDH1a1 has shown high efficiency in catalyzing the oxidation of ACR, 4-HNE, and MDA [118]. Despite this, its primary focus in retinal research has centered on retinal development rather than the detoxification of lipid aldehydes. The reason for this lies in the crucial role of ALDH1a1 in converting retinaldehyde into retinoic acid (RA): a critical signaling molecule for vertebrate eye development [119,120]. Throughout the developmental process, ALDH1a1 is expressed in the dorsal retina [121]. Surprisingly, ALDH1a1 knockout mice display normal retinal structure and function, despite the absence of ALDH1a1 protein in the dorsal neural retina [121]. Nonetheless, the RA produced by ALDH1a1 often serves as a signaling molecule influencing the development of dorsal choroidal vasculature [121]. This process involves the upregulation of Sox9, as a transcription factor, in the dorsal retinal pigment epithelium (RPE) cells. The increased expression of Sox9 leads to the enhanced secretion of VEGF from RPE cells, which guides the development of dorsal choroidal vasculature. 

In the adult rat retina, ALDH1a1 is primarily found in Müller cells, although some staining for ALDH1a1 could also be observed in neurons of the INL [117]. In the context of diabetes, there is a significant decrease in ALDH1a1 at both the mRNA transcript and protein levels [117]. Furthermore, the combined activity of ALDH1a1/ALDH2 is considerably lower in retinal lysates from diabetic rats compared to non-diabetic controls [117]. Previous studies conducted by other researchers have demonstrated that reducing the expression of ALDH1a1 in HEP-1c1c7 cells increases their susceptibility to the accumulation of FDP-Lys when exposed to ACR [118]. This decrease in ALDH1a1 also leads to ACR-mediated cell death, as indicated by caspase 3 activation [118]. Similarly, in primary Müller cell cultures, we have established a direct link between decreased ALDH1a1 activity and the accumulation of FDP-Lys using the ALDH1a1 inhibitor, NCT-501 [117]. The treatment of these cells with this drug results in the increased production of FDP-lysine and decreased cell viability [117]. These findings suggest that the downregulation of ALDH1a1 in the diabetic retina may play a major role in the accumulation of FDP-lysine in DR. The precise reasons for the downregulation of ALDH1a1 in the diabetic retina are still unknown. However, it is worth noting that the expression of ALDH1a1 is regulated by the Nrf2 transcription factor [122], which is activated in response to oxidative stress [123]. Although diabetes enhances oxidative stress in the retina, which would be expected to promote the translocation of Nrf2 into the nucleus, there is a reduction in Nrf2 nuclear localization in both the retina of STZ-induced diabetic rats and in cultured retinal cells under hyperglycemic conditions [124,125]. These observations could explain the downregulation of ALDH1a1 at the mRNA and protein levels, as well as the decrease in ALDH enzyme activity observed in retinal extracts from diabetic rats [117].

### 3.2. ALDH2

ALDH2 is a mitochondrial enzyme that is predominantly known for its function as the second enzyme in the oxidative pathway that metabolizes alcohol, converting acetaldehyde into acetic acid [126]. However, like ALDH1a1, it also exhibits the ability to oxidize lipid aldehydes such as ACR, 4-HNE, and MDA, but with a higher catalytic efficiency [127,128]. However, research has indicated that ALDH2 can be inactivated by both ACR and 4-HNE at concentrations >10 µM [127], suggesting that alternative ALDH enzymes might be better suited for manipulation in therapeutic approaches targeting lipid aldehyde detoxification.

Immunohistochemical investigations have revealed significant expression of ALDH2 in all layers of the rat retina [117]. Notably, we have shown that it mainly co-localizes with CRALBP, aligning with the distinctive pattern of Müller cell processes in the inner plexiform layer (IPL) and along the periphery of the inner limiting membrane (ILM) [117]. Previous research has indicated that the expression of ALDH2 in the retina diminishes with age [129]. Furthermore, transgenic mice with increased levels of ALDH2 display protection against retinal damage associated with aging [129]. This protective effect is possibly due to mechanisms that involve reduced apoptosis induced by oxidative stress and an enhanced unfolded protein response in the ER [129]. ALDH2 has attracted some attention concerning its involvement in DR, particularly due to a specific genetic polymorphism prevalent in East Asian populations. This polymorphism leads to diminished enzyme activity [130]. Termed ALDH2*2, this variant allele results from a single point mutation (G to A) in exon 12, resulting in an amino acid substitution from glutamine to lysine (E487K). Apart from causing alcohol-induced “flushing” syndrome, this genetic defect impairs the detoxification of lipid aldehydes [131]. Importantly, the ALDH2*2 allele has been identified as an independent risk factor for the development of DR [132]. However, this association is significant only among individuals who consume alcohol and not among non-drinkers [132]. Besides epidemiological studies, pre-clinical research has indicated the crucial role of ALDH2 activation in alleviating early-stage diabetic retinal damage through upregulation of sirtuin 1 (SIRT1) [133]: a protein known to mitigate inflammation and oxidative stress while also enhancing mitochondrial function [134].

### 3.3. ALDH3a1 

Several studies have shown that ALDH3a1 has a strong affinity for 4-HNE; however, it has a limited ability to detoxify MDA and ACR [127]. The research on ALDH3a1’s role in detoxifying lipid aldehydes has mainly focused on its function in the cornea, where it serves as a crystallin protein and constitutes approximately 30% of the soluble protein content in mammalian corneas [135,136]. Its abundant presence in the cornea is believed to be crucial for maintaining transparency because excessive exposure to UV radiation can induce lipid peroxidation in corneal cells, obstructing the passage of light [137]. In fact, the up-regulation of ALDH3a1 has been shown to protect human cultured corneal epithelial cells from UV exposure and oxidative damage caused by 4-HNE in laboratory experiments [138]. Regarding diabetes and DR, studies utilizing an Aldh3a1 knockout zebrafish model have revealed the impaired detoxification of 4-HNE after Aldh3a1 knockout [139]. This leads to increased systemic levels of 4-HNE, triggering pancreatic damage, hyperglycemia, and retinal vascular alterations similar to early-stage DR [139]. Additionally, ALDH3a1 has been investigated in relation to the neuroprotective mechanisms of TGF-β in preventing hyperglycemia-induced RGC death. The results indicate that TGF-β shields RGCs from hyperglycemia-induced damage through the upregulation of ALDH3a1, along with other antioxidant and stress response proteins like HO-1, Nrf2, and HIF-1α [140]. 

### 3.4. ALDH3a2

ALDH3a2 is primarily associated with Sjögren–Larsson syndrome (SLS): a condition characterized by impaired fat metabolism and skin abnormalities [141]. SLS is caused by mutations in the ALDH3a2 gene, resulting in deficient enzyme activity. Individuals with SLS typically develop unique retinopathy during mid-childhood, exhibiting symptoms such as photophobia, macular dystrophy with perifoveal crystalline inclusions (“glistening white dots”), foveal atrophy with fluid-filled cysts, and mild retinal pigmentary changes [142]. The retinal abnormalities in SLS are likely attributable to the accumulation of lipid aldehydes and/or alcohols; however, further research is needed to identify the specific mechanisms involved. In the diabetic retina, ALDH3a2 is downregulated at the mRNA level [117], while both gene and protein levels are decreased in retinal endothelial cells exposed to high glucose conditions [143]. These changes are potentially linked to the upregulation of a specific long non-coding RNA known as lncZNRD1, which is induced by hyperglycemia [143]. Notably, when lncZNRD1 is silenced, a significant increase in the expression levels of ALDH3a2 is observed in retinal endothelial cells cultured under hyperglycemic conditions [143].

### 3.5. ALDH9a1 

ALDH9a1 exhibits a wide range of substrate specificity for different aldehydes, encompassing diverse compounds such as 4-trimethylaminobutyraldehyde (TMBAL), which serves as a precursor for carnitine [144]. Additionally, ALDH9a1 is involved in metabolizing aminobutyraldehyde, a precursor for GABA [144], as well as 3,4-dihydroxyphenylacetaldehyde, a metabolite of dopamine [144], and betaine aldehyde [144]. To date, there have been no specific studies investigating the functional role of this enzyme in the retina. However, its expression in the diabetic retina seems to remain unaltered [117].

### 3.6. ALDH18a1

ALDH18a1 is an enzyme that converts glutamate into glutamate 5-semialdehyde; however, its specific role in the diabetic retina remains largely unexplored. It is known, however, that DR is characterized by an accumulation of glutamate, which contributes to neurodegenerative changes, as reviewed by [145]. This accumulation can be caused by various mechanisms, such as the inhibition of glial glutamate transporters and the downregulation of glutamine synthase [146]. Notably, ALDH18a1 is markedly downregulated in the diabetic retina, suggesting its potential involvement in these pathological alterations [117].

### 3.7. AKR1b1

Under oxidative stress conditions, AKR1b1 has been shown to contribute to the detoxification of aldehydes such as 4-HNE and its glutathione adducts (GS-HNE) [147]. Specifically, AKR1b1 converts HNE and GS-HNE into 1,4-dihydroxynonene (DHN) and GS-DHN, respectively [148]. However, AKR1b1’s role extends beyond aldehyde metabolism. It also plays a pathological role in diabetic complications by converting excess glucose into sorbitol, although it is a relatively poor catalyst for glucose reduction [149]. It is worth noting that the apparent lack of efficacy and associated toxicity of AKR1b1 inhibitors in clinical trials for diabetic complications could be due to their counteracting effects, leading to the accumulation of lipid peroxidation products [149,150]. This has led to the recognition that AKR1b1’s normal cellular function is more related to aldehyde metabolism than glucose reduction. Moreover, there is evidence suggesting that 4-HNE and GS-HNE, reduced by AKR1b1, can induce NFκB-regulated genes and mediate an inflammatory response that is detrimental to cell function and survival [151]. This implies that AKR1b1 is part of a signaling cascade that triggers inflammation in response to oxidative stress rather than solely acting as a detoxification enzyme. AKR1b1 is predominantly located in Müller glia within the rat retina, and its gene and protein expression levels remain unaltered during experimental diabetes [117]. Due to the conflicting roles of AKR1b1 in glucose metabolism, lipid aldehyde detoxification, and inflammatory signaling, this enzyme might not represent the best target for manipulation in DR. 

### 3.8. AKR1c19

AKR1c19 exhibits unique characteristics compared to other members of the AKR superfamily. Its main function is believed to be as a reductase of xenobiotic α-dicarbonyl compounds, although its catalytic efficiency is significantly reduced when dealing with aldehyde compounds [152]. While AKR1c19 is moderately downregulated in the diabetic rat retina [117], its precise localization and role in this tissue remains unclear. 

### 3.9. AKR7a2

AKR7a2, an alfatoxin A1 dehydrogenase, has been shown to effectively reduce various toxic aldehydes, including ACR and 4-HNE [153]. In individuals with Alzheimer’s disease, AKR7a2 expression is elevated in the cerebral cortex, indicating its potential role to counteract the harmful effects of aldehyde accumulation associated with this condition [154,155]. Within the retina, AKR7a2, like several other enzymes involved in detoxifying lipid aldehydes, is primarily localized in Müller glia cells [117]. Although there is a decrease in AKR7a2 expression in the diabetic retina, the precise implications of this in the development of DR have yet to be fully explored.

## 4. Therapeutic Approaches Relating to Aldehyde Detoxification

Based on current evidence, lipid aldehydes, and ALEs are believed to have a significant role in the development of DR. Therefore, targeting and reducing their impact in the retina could potentially serve as a valuable therapeutic approach. While scavenging agents that remove harmful adducts have shown some success in preclinical models [70,108,156,157], these approaches have not yet entered the clinical realm for the targeted treatment of early-stage DR. The reason for this disappointing translation into clinical care reflects the relative lack of robust biomarkers and early-stage DR progression readouts; thus, the challenge remains in designing clinical trials within a reasonable time frame. Furthermore, scavenging agents often require high systemic doses, leading to reduced specificity and an increased risk of toxic side effects. In the context of the current review, one potential targeted approach involves enhancing the expression and activity of detoxifying enzymes, specifically ALDH and AKR, to mitigate lipoxidation and reduce ALE accumulation during DR. This could be achieved through methods such as gene therapy or the use of pharmacological compounds.

### 4.1. Gene Therapy

The utilization of gene therapy in the eye provides the opportunity for long-lasting expression of therapeutic genes, with the added advantage of localized tissue delivery and the cell-specific expression of the transgene. Beyond the value of gene delivery to establish molecular mechanisms and define potential therapeutic targets, viral vector-based gene therapy has gained considerable traction in ocular applications. These have shown therapeutic efficacy for some rare monogenic disorders, where the successful prevention of retinal degeneration has been achieved [158,159,160,161,162]. 

Adeno-associated viruses (AAV) are the most widely used vectors for gene therapy in the eye [163]. These vectors can be combined with various promoters to achieve active gene expression in specific targeted cells [164]. In the context of disease-associated accumulation of aldehyde adducts, Müller cells present an appealing target for gene therapy. In fact, Klimczak et al. have demonstrated the effective and selective transduction of Müller cells through the intravitreal delivery of AAV [165]. Similarly, the manipulation of Müller cell aquaporin (AQP) levels has been achieved using a similar method [166]. Gene therapy targeting Müller cells has also shown promise in diabetic mice, where modifying the expression of a soluble and potent form of angiopoietin 1 (AAV2.COMP-Ang1) prevented various pathological indicators, including leukocyte-endothelial interaction, retinal oxygenation, vascular density, vascular marker expression, vessel permeability, retinal thickness, inner retinal cellularity, and retinal neurophysiological responses [167]. 

Expanding on these published findings, the prospect of delivering vectors that elevate protective enzyme expression in a cell-specific, safe, and sustained manner to the diabetic retina holds potential as a valuable therapeutic avenue to prevent disease progression. Nevertheless, there are several constraints in the delivery of vectors for early-stage DR treatment. Therefore, it is of the utmost importance to prioritize the progression of AAV development and delivery, allowing for targeted administration to the different cells of the NVU linked to DR. This can be achieved by utilizing modified viral vector capsids, lipid-based nanoparticles, or polymer systems, with the objective of improving transduction efficiency and reducing immunogenicity in the retina [168,169,170,171]. Such therapeutic development also needs to proceed hand in hand with improved phenotyping for DR and the selection of patients with defined retinopathic profiles who are most likely to gain maximum benefit from a gene therapy approach. 

### 4.2. Pharmacological Approaches

Several small molecules have been identified as potential modulators of aldehyde detoxifying enzymes, and their exploration has varied across different disease scenarios. For instance, tamoxifen, an estrogen receptor antagonist employed in breast cancer therapy, exhibits selective binding to the aldehyde site of ALDH1a1. This binding results in an approximately twofold increase in enzymatic activity without affecting human ALDH2 or ALDH3a1 [172]. Similarly, omeprazole, a medication utilized to decrease stomach acid production, is a potent activator of ALDH1a1. It significantly boosts enzymatic activity by 4–6 times, making it the most effective activator identified to date [173]. Alda-1 (N-(1,3-benzodioxol -5-ylmethyl)-2,6-dichlorobenzamide) was the first activator of ALDH2 to be discovered [174]. Notably, Alda-1 has shown significant potential for the treatment of DR. It effectively reduces oxidative stress and inflammation, restores retinal function, and has been shown to safeguard the structural integrity of diabetic mouse retinas [175]. Alda-341 (d-limonene), which is present in citrus peel oil and widely used as a food flavoring, acts as a specific activator of ALDH3a1. Its activation significantly enhances submandibular gland shape and function after radiation treatment, boosting sphere-forming capacity, reducing apoptosis, and decreasing aldehyde accumulation in salivary stem cells and embryonic glands [176]. Another ALDH3a1 activator, Alda-89 (5-allyl-1,3-benzodioxole), protects against radiotherapy-induced salivary gland dysfunction by promoting the survival and proliferation of salivary stem cells in vivo [177,178]. 

In addition to direct enzyme activators, there are also compounds capable of enhancing the expression of aldehyde-detoxifying enzymes. For example, activation of the constitutive androstane receptor by 1,4-bis [2-(3,5-dichloropyridyloxy)]benzene (TCPOBOP) leads to increased mRNA levels of AKR1b7, AKR1c6, AKR1c19, and AKR1d1 [179]. On the other hand, the activation of the pregnane X receptor (PXR) by 5-pregnenolone-16α-carbonitrile enhances AKR1b7 mRNA but inhibits the expression of AKR1c13 and AKR1c20 mRNA [179]. Furthermore, the Nrf2 activator 2-cyano-3,12-dioxooleana-1,9-dien-28-imidazolide (CDDO-Im) promotes the expression of AKR1c6 and AKR1c19 mRNAs [179].

## 5. Conclusions

In conclusion, DR is a progressive disease that significantly threatens vision globally. Specific lipid aldehydes (ACR, 4-HNE, MDA, and GO) have strong associations with the onset and progression of DR. The retina possesses defense mechanisms that detoxify these lipid aldehydes produced during lipid peroxidation. Targeting ALDH and AKR enzymes, which are crucial for aldehyde detoxification, shows immense potential as a therapeutic approach for DR. These enzymes are expressed in the cells of retinal NVU and understanding their expression and functional roles in the retina, particularly in the context of diabetes, is crucial when unraveling the mechanisms underlying lipid aldehyde accumulation and DR development. Further research is needed to fully understand the cell-specific expression of various aldehyde detoxifying enzymes in the retina, which could be accomplished through the utilization of single-cell RNA sequencing techniques. Additionally, a deeper understanding of the substrate specificity of different aldehyde detoxifying enzymes could help identify the most suitable targets to enhance DR treatment. It is now imperative that we prioritize the translation of this research into therapeutic strategies that could be effectively implemented in a clinical setting. 

## Figures and Tables

**Figure 1 antioxidants-12-01466-f001:**
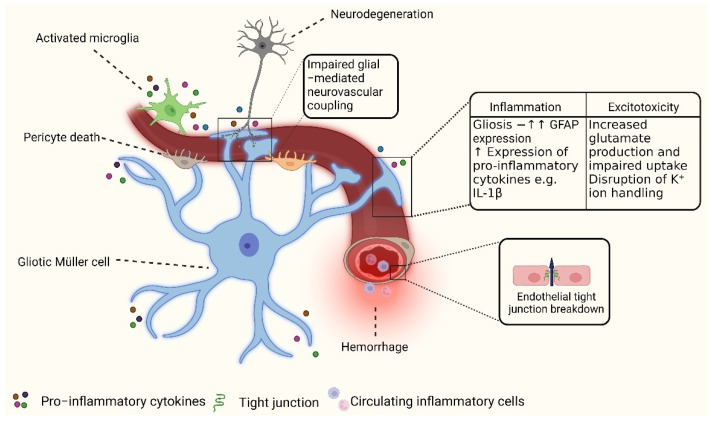
**Mechanisms of neurovascular unit dysfunction in DR.** Under normal conditions, the neurovascular unit (NVU) comprises vascular endothelial cells, pericytes, glial cells, microglia, and neurons, which work together to maintain neurovascular coupling and the integrity of the inner blood-retinal barrier (iBRB). In diabetes, the NVU is disrupted through several mechanisms. Müller cells, crucial for supporting retinal neurons, undergo gliosis and exhibit a pro-inflammatory state. Additionally, they experience impaired regulation of glutamate and potassium levels in the extracellular space. These changes disrupt neurovascular coupling, compromise iBRB integrity, and lead to neuronal apoptosis. In parallel, microglia become pro-inflammatory and contribute to neurovascular degeneration. Increased oxidative stress and inflammation result in pericyte cell death and compromised tight junctions at the iBRB. The breakdown of iBRB leads to hemorrhage and leakage from blood vessels, facilitating the infiltration of pro-inflammatory cells from the circulation.

**Figure 2 antioxidants-12-01466-f002:**
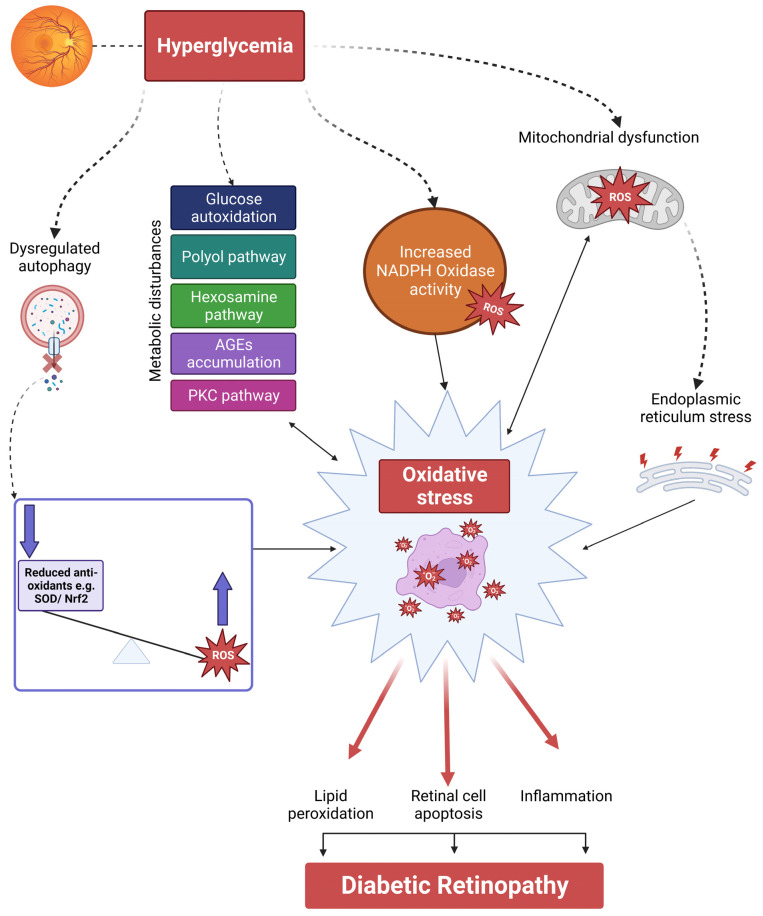
**Pathways of Oxidative Stress in DR.** Retinal hyperglycemia triggers multiple mechanisms that culminate in oxidative stress, as described from left to right in the figure. Dysregulated autophagy disrupts retinal antioxidant mechanisms, leading to an imbalance between the production of ROS and antioxidant activity. Hyperglycemia further enhances ROS production and affects various metabolic pathways in the retina, including the polyol pathway, hexosamine pathway, formation of advanced glycation end products (AGEs), and the protein kinase C (PKC) pathway. These pathways promote oxidative stress, and elevated levels of ROS, in turn, further activating these pathways, creating a detrimental cycle of oxidative stress in DR. Additionally, hyperglycemia contributes to mitochondrial dysfunction and endoplasmic reticulum (ER) stress, which perpetuate oxidative stress in the retina. The high levels of oxidative stress result in increased lipid peroxidation, inflammation, and apoptosis of retinal cells, which are characteristic features of DR.

**Table 1 antioxidants-12-01466-t001:** ALDH and AKR isoforms expressed in the mouse retina.

Gene	Protein Names (Aliases)	Number of Transcripts	Subcellular Location	Chromosome Localization	Substrate Specificity
ALDH1A1	Aldehyde dehydrogenase 1 family member A1	4	Cytosol	9q21.13	Retinal, Acrolein, 4-HNE and MDA
ALDH2	Aldehyde dehydrogenase 2 family member	2	Mitochondria	12q24.2	Acetaldehyde, Acrolein, 4-HNE and MDA
ALDH3A1	Aldehyde dehydrogenase 3 family member A1	10	Cytosol, Nucleus	17p11.2	Aromatic aliphatic, Acrolein. 4-HNE and MDA
ALDH3A2	Aldehyde dehydrogenase 3 family member A2	22	Microsomes, peroxisomes	17p11.2	Fatty aldehydes
ALDH9A1	Aldehyde dehydrogenase 9 family member A1	1	Cytosol	1q23.2	γ-Aminobutyr-aldehyde, 3,4-dihydroxyphenylacetaldehyde, betaine aldehyde
ALDH18A1	Aldehyde dehydrogenase 18 family member A1	2	Mitochondria	10q24.3	Glutamic γ-semi-aldehyde, glutamate
AKR1A1	Aldehyde reductase: dihydrodiol dehydrogenase 1	6	Cytosol	1p33-p32	DL-glyceraldehyde Melvadate
AKR1B1	Aldose reductase	1	Cytosol	7q35	Glucose, advanced end glycation products, 4-HNE, GS-HNE, reactive carbonyls
AKR1B10	Small intestine like aldose reductase; 9-cis-retinal reductase	1	Cytosol and Plasma Membrane	7q33	Retinal; reactive carbonyls
AKR7A2	Aflatoxin aldehyde reductase	3	Cytosol	1p35.1-p36.23	Reduction of succinic semialdehyde; acrolein 2-carboxybenzlaldehyde; aflatoxin dialdehyde, 4-HNE

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
