# Peer review of "Aldehyde Dehydrogenase and Aldo-Keto Reductase Enzymes: Basic Concepts and Emerging Roles in Diabetic Retinopathy"

_antioxidants, 2023, doi:10.3390/antiox12071466_

Round 1

Reviewer 1 Report

This is a great review on aldehyde metabolism and diabetic retinopathy.  The authors covered multiple facets on the pathogenesis of DR.  There is no further comment from this reviewer.

Author Response

This is a great review on aldehyde metabolism and diabetic retinopathy.  The authors covered multiple facets on the pathogenesis of DR.  There is no further comment from this reviewer.

We thank the reviewer for their positive and supportive comments.

Reviewer 2 Report

This is an excellent review. The only comment that I would like to add is on the gene therapy section. I do understand gene therapy is very much talk about since the first ocular gene therapy approval just over 5 years ago. So far, there are no further approval in retinal diseases. For treatment of early stage DR, gene therapy is unlikely to be the answer until probably many years from now. Subretinal gene therapy can only treat a small area and surgically risky. Intravitreal and suparchoroidal gene therapy, so far failed to deliver enough transfection in Muller cells in human / NHP if Muller cell transfection is needed. One additional problem of gene therapy, it lasts forever potentially, so the risk is hard to justify in early stage DR. I would suggest the authors to rewrite that part of the article with a bit more caution. 

Author Response

This is an excellent review. The only comment that I would like to add is on the gene therapy section. I do understand gene therapy is very much talk about since the first ocular gene therapy approval just over 5 years ago. So far, there are no further approval in retinal diseases. For treatment of early stage DR, gene therapy is unlikely to be the answer until probably many years from now. Subretinal gene therapy can only treat a small area and surgically risky. Intravitreal and suparchoroidal gene therapy, so far failed to deliver enough transfection in Muller cells in human / NHP if Muller cell transfection is needed. One additional problem of gene therapy, it lasts forever potentially, so the risk is hard to justify in early stage DR. I would suggest the authors to rewrite that part of the article with a bit more caution.

We express our thanks to the reviewer for their positive and encouraging feedback. We concur that the utilization of gene therapy as a potential treatment for conditions like DR presents a number of challenges. Nonetheless, we believe it is a promising avenue to explore for future advancements in DR treatment. In light of the reviewer's comments, we have incorporated additional details regarding the limitations associated with gene therapy in our revised manuscript (lines 530-535).
